# Roles of *Arbuscular mycorrhizal* Fungi as a Biocontrol Agent in the Control of Plant Diseases

**DOI:** 10.3390/microorganisms10071266

**Published:** 2022-06-22

**Authors:** Wenfeng Weng, Jun Yan, Meiliang Zhou, Xin Yao, Aning Gao, Chao Ma, Jianping Cheng, Jingjun Ruan

**Affiliations:** 1College of Agriculture, Guizhou University, Guiyang 550025, China; wenfengweng@hotmail.com (W.W.); yaoxinn@hotmail.com (X.Y.); gaoanjing@hotmail.com (A.G.); machao@hotmail.com (C.M.); jpcheng@gzu.edu.cn (J.C.); 2Key Laboratory of Coarse Cereal Processing in Ministry of Agriculture and Rural Affairs, Schools of Food and Biological Engineering, Chengdu University, Chengdu 610106, China; yanjun62@cdu.edu.cn; 3Institute of Crop Sciences, Chinese Academy of Agricultural Sciences, Beijing 100081, China; zhoumeiliang@caas.cn

**Keywords:** *Arbuscular mycorrhizal* fungi, biocontrol, disease resistance, mechanism

## Abstract

*Arbuscular**mycorrhizal* fungi (AMF) are a class of beneficial microorganisms that are widely distributed in soil ecosystems and can form symbionts with 80% of terrestrial higher plants, and improve the nutritional status of plants. The use of AMF as a biocontrol method to antagonize soil-borne pathogens has received increasing interest from phytopathologists and ecologists. In this paper, the mechanisms of resistance to diseases induced by AMF and the application of AMF to plant fungal, bacterial, and nematode diseases have been summarized. This study aimed to enhance the potential use of AMF as a biological control method to prevent plant diseases in the future. Root morphological alteration characteristics were explained, including the influence of AMF on root structure, function, and the regulation of AMF via secondary metabolites. AMF can improve the rhizosphere environment by influencing the physical and chemical proprieties of soil, enhancing the growth of other beneficial microorganisms, and by competing with pathogenic microorganisms. Two microorganism types may compete for the same invasive sites in root systems and regulate nutrition distribution. AMF can induce the host plant to form defense systems, including improving phytohormone concentrations, inducing signal substrate production, gene expression regulation, and enhancing protein production.

## 1. Introduction

With the widespread use of chemical pesticides, fungi, bacteria, nematodes, and other pathogens in farmland ecosystems are becoming increasingly resistant to pesticides, whereas their natural enemies have been killed in large numbers, resulting in increasingly rampant pests and diseases [1]. Simultaneously, chemical pesticides pollute the atmosphere, water, and soil. They can remain in animals and plants, enter the human body through the food chain, and endanger human health [2]. With the continuous improvement in people’s living standards, food safety problems caused by pesticides and fertilizers have received widespread attention [3]. Therefore, the search for green and environmentally friendly technologies to control plant diseases and insect pests has become one of the research hotspots of environmental scientists and plant pathologists [4]. Biological control is technology that has received considerable attention [5]. It has been widely studied by soil scientists, plant pathologists, and ecologists owing its high efficiency, low consumption, environmental safety, and diverse functions [6,7,8]. Presently, major breakthroughs in the use of parasitic and predatory insects (natural enemies) to control diseases and insect pests have been made, and related technologies have been popularized and applied [9]. Significant progress has been made in the use of microorganisms for biological control.

*Arbuscular mycorrhizal* fungi (AMF) are the most common, largest biomass and the most significant beneficial fungal group in the mycorrhiza, with a specific antagonistic or inhibitory effect on soil-borne pathogens [10,11]. Studies have shown that AMF can regulate the formation of secondary metabolites in host plants by changing the morphology or anatomical structure of plant roots, improving the physical and chemical properties of the rhizosphere environment, competing with pathogens for photosynthetic products and infection space, and activating disease resistance and defense systems in plants [12]. AMF can reduce the damage caused by fungi, nematodes, bacteria, and other pathogens of *Cucumis sativus*, *Fragaria ananassa*, *Lycopersicon esculentum*, *Citrus reticulata*, *Olea europaea*, *Medicago truncatula*, *Cucumis melo*, *Zea mays*, *Solanum tuberosum*, *Musa nana*, and other plants, and can also reduce the use of pesticides [3,4,6,7,10,13]. Biological control is technology that has received a great deal of attention [14]. More than 30 AMF species have proven effective in controlling plant soil-borne diseases [15,16]. Therefore, it is of great theoretical and practical value to study the mechanism of AMF in the biological control of plant diseases. Therefore, this paper reviews the recent applications and mechanisms of AMF in biological control to enhance the disease resistance of host plants.

## 2. Biocontrol of AMF against Phytopathogenic Fungi

Since AMF live in the soil and infect the roots of plants, they have the most significant effect on soil-borne diseases [17,18]. AMF has been widely used as a biological control method to control various phytopathogenic fungi [19]. For example, Schönbeck and Dehne [20] found that mycorrhizal cotton plants were more resistant to infection with the pathogen *Thielaviopsis basicola* than those with sterile roots. Subsequent studies have reported that the chlamydospore yield of *Thielaviopsis basicola* was inversely proportional to the degree of mycorrhizal infection [21]. Chou et al. [22] studied the interaction between AMF and Rhizobium and two pathogenic fungi—*Pythium ultimum* and *Phytophthora megasper*—and found that the presence of mycorrhizal fungi reduced the number of plant deaths caused by *P. megasper*. Compared with the control without AMF inoculation, the disease index and incidence of *Ralstonia solanacearum* inoculated with *G. rhizogenes* and *G. mossie* were reduced by 9.7% and 49.8%, respectively [23]. *G. asciculatum*, *G. etunicatum*, *G. macrocarpum*, *G. Margarita*, *G. heterogama,* and *G. calospora* in AMF can reduce the diseases caused by pathogenic fungi of the genera *Pythium*, *Phytophthora*, *Fusarium*, *Rhizoctonia*, *Macrophomina*, *Pyrenochaeta*, *Thielaviopsis*, *Phoma*, *Cylindrocarpum*, *Ophiobolus*, and *Sclerotium* in barley, peanut, soybean, banana, cotton, kidney bean, onion, tobacco, citrus, peach, poplar, strawberries, red clover, and ginseng [24,25,26,27,28,29]. Sudhasha et al. [30] found that *G. intraradices* inhibited the growth of the pathogenic fungus *F. oxysporum* and proposed that the chemical balance of mycorrhizae inhibited the growth and reproduction of pathogenic fungi. Slezack et al. [31] infected peas with *Aphanomyces euteiches* and found that establishing full AMF symbiosis is essential to plant defense against pathogens. Phytophthora is a model pathogenic fungus that has been widely used for AMF-mediated plant disease control [32]. Steinkellner et al. [25] tested tomatoes with *G. intraradices* and the pathogen *F. oxysporum* and showed that the severity of the disease could be reduced using a combination of phosphorus application and AM fungal pre-treatment. When AMF is used for plant disease control, its control effect is affected by factors such as the type of plant disease, the relationship between AMF and host plants, the amount and time of AMF inoculation, and environmental factors.

## 3. Biocontrol of AMF against Phytopathogenic Bacteria and Nematodes

Weaver et al. [33] found that mycorrhizal inoculation could reduce the damage of bacterial wilt of tomatoes caused by *P. solanacearum*, a worldwide soil-borne disease. Kamble et al. [34] found that the incidence of *P. syringaepv* was significantly reduced by phosphorus application and mulberry inoculation with *G. fasciculatum* or *G. mosseae*. In the grapes inoculated with AMF, the *Pseudomonas fluorescens* on the root surface was reduced, thereby reducing the chance of re-sickness of the grape plants [15]. Miransari [16] found that *G. mosseae* prevented soybean plant infection by suppressing the number of the pathogen *P. syringae* in soybean. The colonization of AMF on the roots of apple seedlings reduced the infection of the roots of apple seedlings by actinomycetes, thereby reducing the disease of apple seedlings [15]. Many studies have reported that AMF can alleviate nematodes diseases. Pre-inoculation of tobacco with *G. mosy* can significantly reduce the number of *T. basicola* and *Meloidogyne incognita propagules* and improve the disease resistance of plants to pathogenic nematodes [35]. AMF can parasitize cysts of soybean cyst nematodes and reduce various nematode diseases of oats, soybeans, cotton, cucumbers, tomatoes, kidney beans, alfalfa, citrus, and peach to varying degrees [36,37]. Shrinkhala et al. [38] proposed that AMF symbionts can improve plant tolerance to nematodes but cannot fundamentally reduce nematode damage. At high concentrations, this effect is inhibitory [38]. Vos et al. [39,40] reported the effect of AMF on tomato root-knot nematodes; after inoculation, the roots of mycorrhizated plants had fewer galls than the control and significantly reduced the infection rate of nematodes. There was a specific inhibitory effect, and half of the larvae died after one day of culture [35]. Vos et al. [40] conducted potted and field experiments to study the inhibitory effect of AMF on *M. incognita*. The results showed that AMF promoted the growth of plants damaged by *M. rhizogenes* and reduced the root-knot index [40]. Different AMFs have specific control effects on *M. rhizogenes*. Extremely complex interactions between AMF, plants, pathogenic bacteria, and other microorganisms exist due to the various materials used by different researchers or different experimental conditions. Therefore, AMF disease resistance in plants is not consistent [39].

## 4. AMF and Beneficial Microorganisms Combine to Control Plant Diseases

The combined inoculation of AMF and Trichoderma was better than the single inoculation, and the incidence and severity of plant diseases significantly reduced. The study found that the combined inoculation increased the aboveground biomass of *Solanum lycoperdicum* by 11.6–69.7% compared with the single inoculation of AMF mixed inoculum or *T. harzianum*. The combination of the two fungicides has an obvious synergistic effect on the control of plant diseases. Tanwar et al. [41] found that the incidence rate of tomato was 70.0% after inoculation with *F. oxysporum*, the incidence rate decreased to 20.0% after inoculation with *G. mosseae* and *Acaulospora laevis*, and the incidence rate of one AMF combined with *T. virid* was only 10.0%. The combined inoculation of two AMF and Trichoderma can completely inhibit disease occurrence. However, the control effects of different combinations of AMF and Trichoderma on plant diseases showed certain differences. Martinez-Medina et al. [42] studied under field conditions and found that, compared with the control, after inoculation with AMF, there was a 25.0% to 50.0% reduction in incidence of *Cucumis melo* fusarium wilt, and after inoculation with *T. harzianum*, the incidence rate of plants reduced by 60.0%. After mixed inoculation, other than *G. claroideum* + *T. harzianum*, the other combinations reduced the incidence rate by more than 60.0%. Not only that; the combination of the same AMF and Trichoderma has different effects on different varieties of the same species. Sennoi et al. [43] studied the control effect of *G. clarum* and *T. harzianum* on *Halianthus tuberosus* and found that the best control effect for variety HEL246 was the combined inoculation of Trichoderma and AMF, while the best control effect for variety JA37 was AMF alone.

The combined use of AMF and Pseudomonas has an obvious synergistic effect on improving plant disease resistance, and its control effect is better than that of single inoculation treatment. Singh [44] found that single inoculation of *G. sinosum*, *G. albida* or *P. fluorescen* reduced the incidence rate of *Phaseolus vulgaris* by 50.5–52.8%, combined inoculation reduced the incidence rate by 68.9–69.2%, and the nitrogen and phosphorus contents of the plants treated by combined inoculation were higher than those treated by single inoculation. The combined inoculation of *G. sinuosum* and *P. fluorescen* was also more effective than inoculation alone against tomato fusarium wilt [45,46] and *Carica papaya* root rot [47] caused by *F. oxysporum*. However, the effects of co-inoculation of AMF and *P. fluorescen* on plants were not necessarily all positive. Behn [48] found that in the absence of pathogenic microorganisms, the growth promotion effect of *G. messeae*+*P. fluorescen* was better than that of single inoculation, but the control effect of single inoculation on pathogens was better than that of combined inoculation. In addition to *P. fluorescens*, the combined inoculation of *P. aeruginosa* and AMF also had better biocontrol effects. Inoculation with two *G. intraradice* and *G. clarum* alone can reduce the severity of *Elaeis guineensis* base rot by 15.0–17.0%, and combined inoculation of two AMFs with *P. aeruginosa* can reduce the severity by 57.0–80.0% [49].

The combined treatment of AMF and Bacillus has an excellent control effect on plant root diseases. Mixed inoculation of *G. mosseae* and *Bacillus subtilis* can not only reduce the severity of tomato fusarium root rot by 85.0–93.4%, but can also improve plant nutrients (nitrogen, phosphorus, potassium, calcium, magnesium, iron, and zinc), leaf pigment, total soluble sugar, total soluble protein, and total free amino acid content [50,51]. Zhang et al. [52] found that *G. versiforme* and *B. vallismortis* inoculation alone reduced the disease index of cotton verticillium wilt by 37.7% and 35.7%, respectively, and combined inoculation of the two reduced the disease index by 63.3%. The combined inoculation of AMF and Bacillus has a control effect of 73.6–82.1% against *F. oxysporum* [51], while the control effect of single inoculation is only 34.1–52.1%. In addition, under greenhouse conditions, the control effect of combined inoculation of *G. mosseae*, *G. monosporum*, *A. laevis*, *R. clarus*, and *B. amyloliquefaciens* on *Allium sativum* white rot [53] is also better than that of single inoculation. They can not only promote the growth of garlic and increase the yield, but also increase the contents of chlorophyll a, chlorophyll b, and carotenoids in the host. However, the control effects of the combination of AMF and Bacillus on different diseases were significantly different. Glomus can improve the control effect of *B. subtilis* on strawberry fusarium wilt [54]. Compared with single inoculation, combined inoculation increases the fresh weight of strawberries by 61.7–90.9%. However, *G. mosseae*, *G. caledonium*, and *G. sinuosum* will affect the control effect of *B. subtilis* on strawberry *powdery mildew* [55]. Inoculation with *B. subtilis* alone can reduce the relative severity of strawberry *powdery mildew* by 56.0%, and only 12.0% after combined inoculation with AMF.

## 5. The Mechanism of AMF Improving Plant Disease Resistance

Currently, research on the use of AMF for plant disease control is increasing. Analysis of its related mechanisms mainly includes improving plant nutrition, changing plant root morphological structure, regulating the synthesis of secondary metabolites, improving plant rhizosphere microenvironment, directly competing with pathogenic microorganisms for invasion sites and nutrients, and inducing plant disease resistance and defense system formation (Figure 1) [56,57,58,59,60,61].

### 5.1. AMF Regulate Plant Root Morphological Structure

Pathogens infecting plant roots must enter cells through the cell wall, and AMF symbiosis can cause the root system of the host plant to grow, thicken, and increase branching, accelerate the lignification of the cell wall, thicken the root tip epidermis, and increase the number of cell layers; it can also change the root morphological structure, thereby effectively slowing down the process of roots infection by pathogens [62,63] (Figure 2A). Under the stress of *Verticillium dahliae*, the root xylem structure of *Gossypium hirsutum* symbiotic with *G. mosseae* and *G. etunicatum* increased, palisade tissue and vessels were deformed, gelatinous substances were produced at the vessels, the cells deformed and shrank, the color deepened, and the cell walls significantly thickened. Lignification, a significant reduction in the number of vacuoles, the disappearance of mitochondrial inner folds, and a series of structural changes in the root system are all beneficial for improving the resistance of host plants to *Verticillium dahliae* [63].

AMF can form a mycelium network, callose, and papillary structure stacked by non-esterified pectin in the root epidermis and endodermis of host plants (Figure 2B), which hinders pathogen penetration into root cell tissues and further infection, changes in tomato root anatomy alter the infection kinetics of *Phytophthora infection* [64]. Changes in the root systems of mycorrhizated plants are also reflected in the induction of cell walls to produce hydroxyproline-rich glycoprotein (HRGPs) [65]. HRGP is a linear compound molecule of sugar protein on the plant cell wall. As a cell wall structural substance, it can improve the strength of the host plant cell wall, so that it cannot be decomposed by proteases, cellulases, hemicellulases, secreted during pathogen infection. Conversely, HRGP acts as a lectin in pathogenic invasion of plants and fixing of pathogens on the cell wall (Figure 2C), thereby preventing them from invading plant cells.

In addition to affecting the root structure of host plants, some substances present on the AMF cell wall can also inhibit pathogenic bacteria (Figure 2C). Studies have shown that β-1, 3-glucan is present on some *Glomus* AFM extra-root hyphae, the inner wall of germ tubes and spore cell walls, but β-1, 3-glucan does not exist in *Scutellospora* or *Gigaspora* fungi [66,67]. β-1, 3-glucan is a structural component of the cell wall, and its presence in AMF indicates that AMF has a barrier effect on pathogenic microorganisms. AMF can enhance the resistance of host plants to pests and diseases by changing the anatomical structure of plant roots and spontaneous structural substances to achieve biological control [68,69].

### 5.2. AMF Improves Plant Nutrition

AMF improved the uptake of mineral nutrients and water by host plants. The study found that the AMF extra-root mycelia widely distributed in the soil intertwined to form a huge mycelium network, thereby expanding the absorption range of the root system for water and nutrients, especially phosphate and nitrate, and for different plants at the same time. Water and nutrients are redistributed so that plants have another effective water and nutrient passage to a certain extent [70]. In 1993, Eissenstatet et al. [71] used 14C labeling technology to detect mycorrhizal citrus roots and found that very little 14C was released around the roots. Therefore, there must be competition for photosynthetic products among the microorganisms around the roots. AMF and pathogens compete for photosynthetic products from the root system of the host plant, and when photosynthetic products are first utilized by AMF, the opportunity for pathogen acquisition is reduced, thereby limiting the growth and reproduction of pathogens [72]. Mycorrhizae compensate the loss of root biomass and function caused by pathogenic infection by enhancing the absorption of nutrients and water by plants, thereby indirectly reducing the damage caused by pathogenic microorganisms and improving the disease resistance of host plants [73]. Inoculation of tomatoes under *F. oxysporum* stress can promote the absorption of K, N, P, Ca, Mn, Zn, and other elements), increase chlorophyll and soluble sugar content, and increase the number of branches and leaves of the plant. Enhanced biological activity indirectly improves the plant’s tolerance to pests and diseases [74,75,76]. A study found that the growth of olives damaged by *M. incognita* and *M. javanica* significantly improved after inoculation with AMF, and the biomass increased by 88.9% [77].

### 5.3. AMF Regulate the Synthesis of Plant Secondary Metabolites

Mycorrhizal symbionts can regulate the types and quantities of secondary metabolites in the physiological metabolism of host plants, an important mechanism by which AMF induce plant disease resistance [77,78]. AMF can produce chemical substances such as phytochemicals, calloses, alkaloids, and phenols on the surface of both the inner and outer hyphae of the root, and these secondary metabolites are beneficial for plants, helping them to resist adverse conditions caused by diseases [79]. Phytoprotectins are a class of resistant compounds produced during plant infection by pathogenic microorganisms, and their production speed and accumulation are related to disease resistance in plants [79]. Phytophanins accumulate around infected cells and act as barriers to prevent the further spread of pathogens [80]. *G. mosseae* can induce plant roots to produce a phytotoxin stress response and improve plant disease resistance [78]. Inoculation with *G. intraradices* can promote callose deposition in the roots and protect cucumbers from *Colletotrichum orbiculare* poisoning [81]. The vinblastine content in the leaves of *Catharanthus roseus* significantly increased after AMF infection, which improved the plant’s resistance to biotic stress [82]. The phenolic substance family includes various compounds such as flavonoids and phenolic carboxylic acids, which are all related to the signaling molecules and defense systems of plants [79]. Flavonoids can also increase the invasion sites of AMF and increase its infection rate [83]. Studies have found that the content of these phenolic substances in the roots of mycorrhizal plants changes, and a large number of phenolic compounds accumulate in the *Gossypium hirsutum* symbiotic with AMF, which can improve the ability of the host plant to resist *Verticillium dahliae* [84]. The disease incidence and severity of *F. oxysporum* and *C. gloeosporioides* in the aerial parts and roots of strawberries inoculated with *G. mosseae* were significantly reduced, and the total polyphenol and ascorbic acid contents of the plant increased [85]. Zhu et al. [86] found that G. versiforme could induce systemic resistance to Ralstonia solanacearum in tomato plants through root-splitting experiments, and the content of phenolic substances in its roots increased significantly. An increase in phenolic substances occurred in both the infected and uninfected root segments of *R. solanacearum* [79]; therefore, plant system resistance was closely related to the increase in phenolic content. However, some studies found no significant change in phenolic compounds between AMF-vaccinated and AMF-unvaccinated *Phoenix dactylifera F. oxysporum* infection. Mycorrhizated plants accumulated a large amount of hydroxycinnamic acid derivatives, which improved their ability to resist chlorosis in date palms [80].

### 5.4. Regulation of Microbial Populations in the Rhizosphere of Plants

After AMF form a symbiotic relationship with plants, the development of fungal mycelia can change the permeability of the host plant root cell membrane, change the composition and quantity of root exudates, affect the physical and chemical properties of the rhizosphere soil, and further lead to changes in the population structure and quantity of rhizosphere microorganisms (Figure 2D–F). Studies have shown that mycorrhizae and their extra-root hyphae can penetrate through the tiny pores between soil particles, and their secretions, such as glomus associated protein (GRSP), organic acids, and polyamines, can be used as adsorption agents for the adhesion of soil particles, promote the formation of soil aggregate structures, improve soil pH, water stability, aeration, and water permeability, further increase the redox potential (Eh), and promote normal plant growth to resist disease invasion [33,61,87]. The exudates produced by plant roots symbiotic with AMF can directly affect the growth, development, and reproduction of other soil fungi [88,89,90], nematodes [39], and bacteria [33], making the microbial community in the soil in terms of structure, nature and quantity, and variation in spatial distribution. Mycorrhizal tomato root exudate can continue to paralyze nematodes, thereby reducing the invasion of nematodes into the root system [35], and repels the zoospores of *Phytophthora nicotianae*, rendering them unable to access plant roots [84]. AMF can also promote a relationship with beneficial soil microorganisms, resulting in a synergistic effect. This particularly stimulates the activity of microorganisms that are antagonistic to soil-borne pathogens, increasing the number of beneficial microorganisms in the rhizosphere, such as *Gliocladium*, *Streptomyces*, *Trichoderma*, other fungi, nitrogen-fixing bacteria, phosphorus-dissolving bacteria, *Pseudomonas fluorescence*, Bacillus, and other plant-growth-promoting Rhinoacteria (PGPR) and Actinomycetes [85,87]. These beneficial microorganisms also reduce the number of pathogens and the chance of pathogenic microorganisms infecting plants, thereby indirectly improving the resistance of plants to pathogens. PGPR can also strengthen the symbiotic relationship between AMF and plants [88].

### 5.5. AMF Compete with Pathogenic Microorganisms for Invasion Sites

As biotrophic symbiotic microorganisms in the soil rhizosphere, AMF often have the same ecological niche and invasion site as soil-borne pathogens. Therefore, under natural habitat conditions, AMF and pathogens must have a spatially competitive relationship, and their biocontrol effect is mainly to reduce the initial infection and re-infection of root epidermal pathogens. Cordier et al. [88] found that *Phytophthora nicotianae* was reduced in *G. moshe* infection and adjacent uninfected root systems, and the pathogen did not invade arbuscular cells. AMF-infected roots and adjacent uninfected roots in mycorrhizated plants had very little *Heterodera glycines* [89]. Tabin et al. [90] inoculated AMF and pathogenic bacteria in different orders of *Aquilaria agallocha*, confirming competition for infection sites between AMF and pathogenic bacteria. When *G. fasciculatum* was first inoculated, the root site was infected with AMF, which could significantly inhibit the development of *Pythium aphanidermatum* in the root tissue of *Aquilaria agallocha* and reduce the morbidity index and damping-off symptoms of the plant [91]. In the galls of nematodes, vesicles, hyphae, and even arbuscular invasion, AMF can often be seen forming the parasitic effect of AMF on nematodes. Castillo et al. [92] reported that *G. polygamyces* could infect the eggs of soybean cyst nematode. Vos et al. [40] found that cysts of soybean cyst nematodes were colonized with chlamydospores of AMF, which also indicated that AMF had a parasitic effect on the pathogen. However, the parasitic relationship between AMF with pathogenic microorganisms and their biocontrol effect needs further study.

### 5.6. AMF Activate the Host Defense System

#### 5.6.1. AMF Promote the Synthesis of Plant Defense Hormones

Plant hormones, as trace signal molecules in plants, play an important role in regulating plant growth, development, and response to the environment. During AMF growth and establishment of symbiotic relationships with host plants, they can directly synthesize or induce the production of hormone, such as auxin (IAA), cytokinin (CK), giberellin acid (GA), brassinosteroids (BR), jasmonic acid (JA), salicylic acid (SA), ethylene (ET), abscisic acid (ABA), which are also involved in the establishment of AMF-induced disease defense systems in host plants [93]. The study found that cucumber plants inoculated with *G. terrestris* had higher contents of IAA, GA, and zeatin. The relative increase in the content of growth-promoting substances was related to the enhancement of the plant’s disease resistance to *Rhizoctonia solanacearum*. IAA plays a role in the defense process of plants against pathogenic infection [94]. Ortu et al. [95] found that GA synthesis genes were upregulated in *Medicago truncatula* after AMF infection. The contents of endogenous hormones such as IAA, GA, ET, CK, and ABA in the stems and leaves in mycorrhizated plants were higher than those of uninoculated plants. AMF promote plant growth and indirectly enhance disease resistance by affecting the content of endogenous plant hormones and the balance between them. Plant hormones may promote gene expression under stress, which can induce the expression of many new genes and protein synthesis [88].

#### 5.6.2. AMF Induce the Synthesis of Plant Signal Substances

After symbiosis with plants, AMF can induce the synthesis of various signaling substances such as nitric oxide (NO), JA, SA, ET, hydrogen peroxide (H_2_O_2_), ABA, Ca^2+^ signal, and sugar signal [93]. These signaling substances play an important role in the recognition of plants and AMF, the establishment of mycorrhizal symbionts, and activation of plant defense systems [94]. Studies have shown that JA and ET are generally resistant to saprophytic pathogens, whereas SA has an inhibitory effect on biotrophic pathogens. JA and ET are associated with plant systemic induced resistance (ISR), and SA with plant systemic acquired resistance (SAR) when plants are infected by pathogens [96]. Recently, NO has become a hot topic in plant cell signaling research. As a signaling molecule, NO is involved in signaling and gene expression processes related to plant defense systems [97]. The accumulation of NO in plants is closely related to the symbiosis of AMF, and the content of NO in leaves and roots of alfalfa truncates inoculated with *G. margarita* was 3.3 and 1.9 times that of the control treatment, respectively, indicating that AMF can induce the accumulation of NO associated with systemic resistance [98]. Mycorrhizal-induced JA can activate the defense system in plants, thereby increasing plant resistance to pathogens [94]. The roots of the tomato seedlings were infected with *F. oxysporum* for 20 days after inoculation with *G. polyphylla* and *G. macrocarpum*. The disease severity was reduced by 78% and 75%, respectively [25]. SA has been recognized as the main signal substance for AMF to induce systemic disease resistance in plants [99]. Cordier et al. [88] found that inoculation with *G. moses* and external application of SA can effectively inhibit the infection of *F. oxysporum* to tomatoes and reduce the disease index and the degree of wilting. *G. rhizogenes* can increase the phytohormone content of cantaloupe, which is reduced by *F. oxysporum* infection; induce JA and SA signaling pathways; and enhance the defense system of cantaloupe against *F. oxysporum* [25]. However, some studies also found that the endogenous JA and SA contents of *Nicotiana attenuata* plants inoculated with *G. intraradiculae* did not change significantly, and the ET release decreased slightly [100,101]. There are differences in the induction of signaling substances by different AMF species in different host plants, and the related mechanisms need further studies.

#### 5.6.3. AMF Regulate Defense Gene Expression

AMF can enhance the disease resistance of host plants by inducing the expression of genes related to plant defense responses, such as *PAL5* and chitinase gene *Chib1* [102,103], or by regulating the expression and specific expression of various disease resistance genes [104]. Studies have shown that *Glycine max* infected with *Heterodera glycines* was inoculated with AMF, and the genes related to chitinase and phenylalanine ammonia-lyase (PAL) in roots were analyzed using Northern blotting and reverse transcription polymerase chain reaction (PCR) techniques [104]. The mRNA expression of *Chib1* and *PAL5* showed that AMF could regulate these disease-resistance genes related to nematode resistance at the transcriptional level, thereby enhancing their expression and expression intensity [105]. Tahiri-Alaoui et al. [106] isolated the gene *Le-MI-13* from *Lycopersicon esculentum* roots. It was symbiotic with *G. mosei*, which encodes a hydrogen acid sequence similar to ubiquitin that mediates protein degradation and opens or closes gene transcription. The transcription of this gene was only found in mycorrhizal tomato roots by Northern blotting, but not in the roots of uninoculated tomatoes, indicating that AMF is an inducer of the specific transcription of this gene and improves plant disease resistance [101]. Through 16000-type single-nucleotide array and real-time quantitative PCR studies, it was found that AMF induced dramatic changes in defense-related gene transcripts in *Medicago truncatula*, resulting in local and systemic defense responses in plants against plant damage by *Xanthomonas campestris* [104]. Two susceptible maize cultivars (Yuenong-9 and Gaoyou-115) pre-inoculated with *G. moshes* were inoculated with *Rhizoctonia solani*, and it was found that the defense response in the mycorrhizated plants was more rapid and intense. *AOS*, *PR2a* and *PAL* genes related to disease resistance and *BX9*, a key gene in the synthesis pathway of 2, 4-dihydroxy-7-methoxy-2h-1, 4-benzoxazin-3 (4h)-one, and DIMBOA, were strongly induced and expressed in two maize leaves [66]. Two different genotypes of non-mycorrhizated *L. esculentum* (wild-type *76R* and mutant *rmc*) had the same expression levels of defense genes in vivo after infection with *Rhizoctonia solani*. When mutant *rmc* plants were inoculated with mycorrhizae, the extracellular PR-1 and intracellular mRNA expression of *GluBAS* and *Chi9* were more intense [107]. Through real-time quantitative PCR analysis, it was found that the reduction of *P. infestans* disease index and disease symptoms in mycorrhizal *Solanum tuberosum* leaves may be related to the induced expression [106]. The key enzymes involved in JA synthesis are produced in the process of establishing a mutualistic symbiotic relationship between AMF and plants, which can effectively catalyze the synthesis of JA. The expression of cDNAs of allene oxide cyclase (AOC), the main synthase of JA in mycorrhizal *Medicago truncatula* roots, was enhanced, and the accumulated endogenous JA content increased [103].

#### 5.6.4. AMF Improve Defense Enzyme Activity and Induce PR Synthesis

In the process of AMF forming symbionts with host plants, it can activate many defensive enzymes, such as polyphenol oxidase (PPO) and peroxidase (POD) involved in the metabolism of phenolic substances; chalcone isomerase (CHI) is involved in the metabolism of phytoalexin, lignin, and flavonoid/isoflavone biosynthesis; Chalcone Synthase (CHS) is involved in flavonoid synthesis; phenylalanine ammonia lyase (PAL) is involved in the metabolism of phenypropanes, and some defensive proteins associated with disease resistance (such as pathogenesis-related protein, PR protein) were also expressed specifically [108,109]. PAL activity can be used as a physiological indicator of disease resistance in plants. When tomato seedlings infected with *F. oxysporum* were inoculated with AMF, PAL activity in stems and leaves was significantly enhanced, which could significantly reduce disease symptoms [110]. Enhanced superoxide dismutase (SOD) activity and 1,1-Diphenyl-2-picrylhydrazyl (DPPH) free-radical-scavenging activity was observed in strawberry plants inoculated with *G. mosy*. The antioxidant capacity of the plant significantly improved, which in turn enhanced the resistance to *G. anthracis* and *F. oxysporum*, and the level of resistance to *F. oxysporum* was higher [25]. The glucanase, chitinase, PR protein and other substances related to allergic reaction in tomato roots infected with *G. mosei* were higher than those in non-mycorrhizated plants [111]. After inoculation with *G. monosporum*, *G. deserticola*, *G. clarum*, and other inoculants, the activity of polyphenol oxidase in date palm trees was significantly enhanced, which could inhibit the occurrence of chlorosis in *Phoenix dactylifera* [100]. Recent studies have found that *G. rhizogenes* can secrete a defense protein, sp7, that can interact with the disease process-related protein transcription factor ERF19 in the nucleus. The expression of *sp7* can alleviate the symptoms of root rot caused by *Magnaporthe oryzae* [111]. The mechanisms by which AMF increase the disease resistances of host plants are multifaceted, which may be the result of a certain mechanism acting alone or by the synergy of multiple mechanisms. The types of disease resistance induced by AMF may be systemic or local [64,90]. The potency of AMF’s potential antagonism against pathogens depends on the relationship between AMF, host plants and pathogens, affected by abiotic factors such as inoculum size, inoculation period, and soil factors (fertility, pH, temperature, and humidity). In farmland ecosystems, agricultural management measures such as farming systems, fertilization management, and pest management also affect the performance of AMF biocontrol [85]. The biological control effect of AMF can only be exerted when various factors are coordinated (Figure 3).

## 6. Conclusions and Future Prospects

Numerous studies have shown that AMF not only promote plant growth and improve plant health, but also enhance plant resistance to biotic and abiotic stresses. Due to differences in test materials, conditions, sampling sites, and measurement time in the study, the results of different researchers are inconsistent, or even contradict each other. There are many possible mechanisms for AMF to improve plant disease resistance, but it is still unclear which mechanisms play a major role, a secondary role, or do not play a role. Biological control technology represents one of the future developments for plant pathology and has broad prospects. However, the use of AMF for biological control is an emerging pest control technology, and many problems arising in the basic theory and practical applications need to be solved urgently.

AMF cannot be industrialized and mass produced through fermentation, and the limitations of their cultivation methods limit their application. Breaking through the technical bottleneck to realize rapid cultivation of AMF is also a future research priority. Attention should be paid to the safety evaluation of microbial biocontrol, particularly the impact of biocontrol microorganisms on the diversity of other microorganisms in the soil. Studies in this area are rare, and the evaluation of soil health in biocontrol should be strengthened. The biocontrol effect of AMF is affected by many biotic and abiotic factors; therefore, the regulatory factors that can maximize the benefit of plants, such as the optimal inoculation period, inoculation dose, ecological conditions, farming methods, and fertilization amount, should be studied in depth to establish a scientific and effective AMF biocontrol effect evaluation standard and provide a theoretical basis for the use of AMF to carry out biocontrol work.

The application of AMF in plant disease control is bound to become a feasible and ecosystem-friendly solution to reduce the occurrence of pathogens and achieve green and sustainable development. In the future, the germplasm resources of AMF should be systematically surveyed in various places, disease-resistant AMF strains suitable for different host plants should be isolated and screened by conventional and unconventional methods, and an AMF gene bank should be established. Extraction and purification antibacterial substances, (such as antibacterial proteins and antibiotics) from disease-resistant AMF strains should be carried out. In-depth study of the action mechanism among AMF, plants and pathogenic microorganisms, and their deep inner connections will help us to correctly understand the disease resistance of mycorrhizae so that they can be successfully used in biological control. An in-depth study of the influence of AMF on various signaling pathways and the correlation between various signaling substances requires further exploration of which signaling substance is induced by AMF first, and which substances are induced in the second and third order. Which of the gene changes are affected by signal-substance-induced production? Future studies will locate and screen out the related resistance genes induced by AMF, and explore the structure, function, and induced expression mechanism of their products.

## Figures and Tables

**Figure 1 microorganisms-10-01266-f001:**
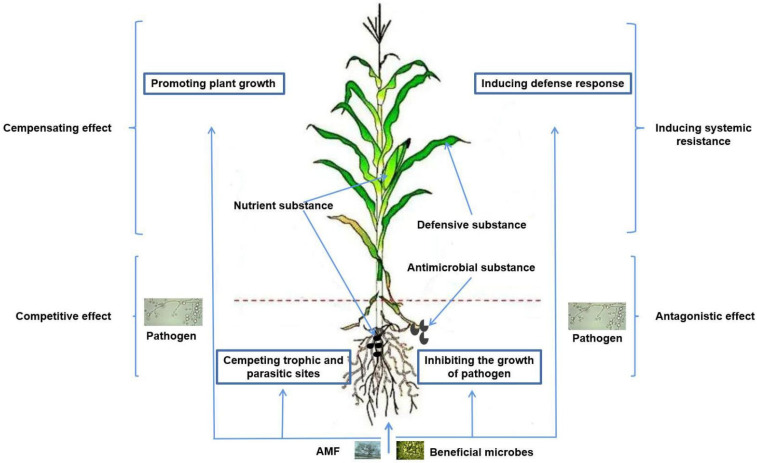
Schematic diagram of the mechanism of joint control of plant diseases by AMF and beneficial microorganisms.

**Figure 2 microorganisms-10-01266-f002:**
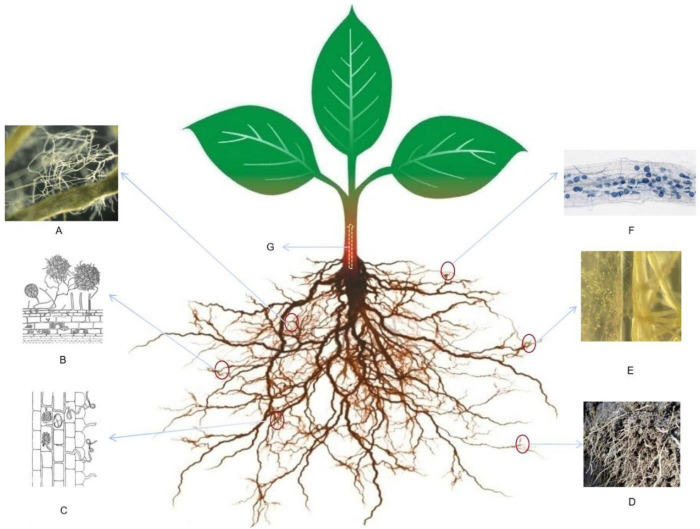
Schematic diagram of biological control mechanisms in arbuscular mycorrhizal fungi (AMF) symbiosis with plants (**A**) Increased root branching and lignified cell walls; (**B**) AMF mycelial network acts as a barrier in root epidermis; (**C**) AMF binds pathogens to the cell wall; (**D**) improve soil structure; (**E**) root exudates kill pathogens; (**F**) AMF stimulates the growth and reproduction of beneficial microorganisms. (**G**) AMF improves the absorption of nutrients and water by plants, while competing with pathogens for nutrients.

**Figure 3 microorganisms-10-01266-f003:**
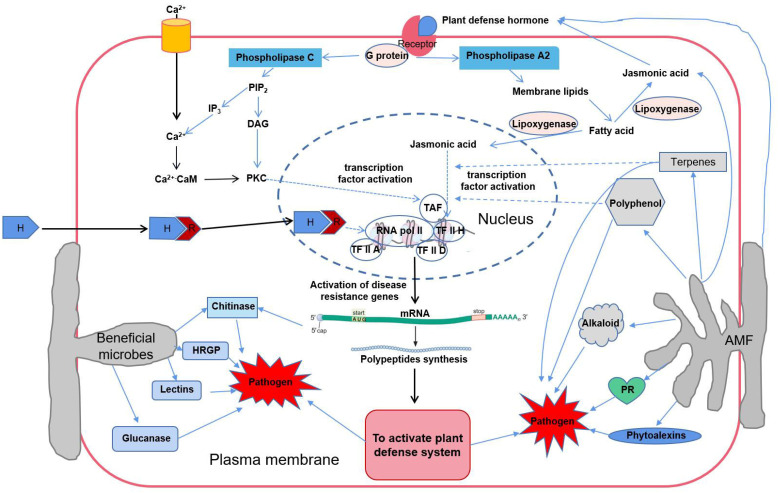
Schematic diagram of the mechanism of action of AMF in activating plant defense systems. Note: PIP_2_—phosphatidylinositol diphosphate; IP_3_—inositol triphosphate; DAG—diacylglycerol; PKC—protein kinase C; H—hormone; R—receptor; HRGP—hydroxyproline-rich glycoprotein; PR—pathogenesis-related proteins.

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
