# Peer review of "Roles of Arbuscular mycorrhizal Fungi as a Biocontrol Agent in the Control of Plant Diseases"

_microorganisms, 2022, doi:10.3390/microorganisms10071266_

Round 1

Reviewer 1 Report

Although the article seems to be properly written, it has to be  improved with more precision. Review articles are beneficial as a source of many references on the subjects gathered on the same place.  My perception is that the authors drew some conclusions that are not supported by proper references or even, some statements are not accurate.  Many times authors give general conclusion about the plant response to mycorrhizal interactions, but in reality, the conclusions of the original paper that the authors referred to, are not so general (depend on many variables such as type of plants, type of pathogen, type of AMF inoculums, additional stimulants like hormones  etc.)

The manuscript is difficult to follow since lots of information is incorporated in the text and the style should be more fluent and less listing-like.

Specific comments:

Line61: More reference should be added to confirm this statement.

Line 175: Reference 56 is not about the resistance of host plants to Verticillium dahliae

Line 178: Reference 57 is not about hindering pathogen penetration into root cell tissues and further infection upom mycorrhization

Line 178: statement that changes in root anatomy alter the infection kinetics of pathogens.. Not sure this is correct so please write kinetics of which pathogen and provide a reference

Line 181: Please provide a reference

Line 238: “Studies have shown that mycorrhizae and their extra-root hyphae can penetrate through the tiny pores between soil particles, and their secretions, such as glomus associated protein (GRSP), organic acids, and polyamines,  can be used as adsorption agents for the adhesion of soil particles, promote the formation  of soil aggregate structures, improve soil pH, water stability, aeration, and water permeability, further increase the redox potential (Eh), and promote normal plant growth to re sist disease invasion [72].”

Please insert more references to confirm your statements

Line 267: “After AMF infestation of plant roots, the number of invasion sites of parasitic pathogens was significantly reduced [73].” Please rewrite and insert which plant and which pathogen

Author Response

(1) Breeding … - first sentence is not clear. I think authors should rewrite complete paragraph to be clear and understandable. What does it mean “… at home and abroad…”? What does it mean “active validity period and stability”? Above all, what does it mean “disease-resistant AMF strains”? As much as I understand, the aim of this point could be to cultivate and construct the bank of AMF strains that cause beneficial processes in plants when plants face pathogenic attack in order to make plant resistance, not to collect AMF strains that are resistant to diseases. This sound strange, maybe even incorrect. Authors also mention possibility of extraction of the antibacterial substances from disease-resistant AMF strains – if such studies are published, this should be described somewhere in the text or authors should explain on what basis do they recommend this. From the whole this paragraph I would keep only the last two sentences in this form.

Also the subtitle “Breeding, domestication and construction of AMF strains with strong disease resistance” is problematic to me – I am not sure that breeding is suitable term (I would rather say cultivation). “Construction of AMF strains with strong disease resistance” sounds like genetic engineering with the goal of production of some super AMF strains. Maybe authors wanted to say “Construction of bank of AMF strains with strong potential to enhance plant hosts’ disease resistance”? – this would be clear to me.

(2) I think authors should write a subtitle or some paragraph in the main-text about combined action of AMF and other beneficial microorganisms. There are numerous studies on this topic published and it is not something that is only a future need.

(4) Throughout the manuscript text authors described numerous examples how AMF may activate some physiological processes that enhance plant immunity system (hormones, enzymes, defence gene expression etc.). And many of such studies refer to above ground plant. So, I don’t see why should “aerial parts of the plants” be mentioned as some future focus when this is already the case. I would omit this.

The entire “Conclusions and prospects” part is insufficiently well written according to me. I suggest writing only main beneficial influences of AMF on control of plant disease and some main future prospects, from which I definitely advise to omit majority of the point (1), moving point (2) to main text, omit point (4) and to shorten point (5) and rewrite it in a clearer way.

We have rewritten this section as requested by the reviewer.

Other comments:

Line 68, 2. Biocontrol of AMF against phytopathogenic fungi – the writing style is mainly listing of what different researchers have found which is difficult to read. I think it should be written more fluid to be easier to follow.

We have rewritten this part.

Line 98, 3. Biocontrol of AMF against phytopathogenic bacteria and nematodes - The same thing about writing style as in subtitle 2. Moreover, too small part refers to AMF influence on pathogenic bacteria.

We have rewritten this part.

Line 126, 4. Plant nutrition – there are numerous papers published on the influence of AMF on plant nutrition in various plant species. This is why I think authors should put much more attention to this topic and write it more complete, not only generally. I suggest to move this subtitle into subsubtitle of the point 5. (The mechanism of AMF improving plant disease resistance). Also, lines 133-138 – do they better suit to the part about competition? Figure 1 – I think this figure is not suitable to the part of AMF influence on plant nutrition, it is not informative enough.

We have rewritten this part.We have omitted Figure 1.

Line 259 – “changing the physical and chemical properties of the plant rhizosphere soil” is not described, even mentioned anywhere in the manuscript. So this can not be conclusion here.

We have removed this sentence.

Lines 275-277 – I don’t think this belongs to this subtitle.

We have removed this sentence.

Line 262 - relation between AMF and viruses is not mentioned anywhere in the manuscript (omit it from the Abstract).

We have omitted it from the Abstract.

Reviewer 2 Report

Authors have written the manuscript dealing with the intriguing topic of arbuscular mycorrhizal fungi roles as biocontrol agents which is under explored and there is indeed a need to lighten it more deeply. Many aspects of AMF roles in enhancing plant resistance are described. However, I have several issues to question:

Title

I did not find in the manuscript text any mention of influence of AMF on insect pests. If this is right, I definitely advise omitting “… and insect pests” from the title.

My major concerns:

I find “Conclusions and future prospects” very unrelated to the manuscript mainbody text. Since complete manuscript describes findings and mechanisms of AMF biocontrol against pathogenic fungi, bacteria and nematodes as well as mechanisms of AMF improvement of plant disease resistance, I find it logical to write few sentences summarizing it or bringing us the main AMF potential as biocontrol agent. I think this is lacking in Conclusions.

Authors gave effort to suggest which research aspects in this context should be strengthened in future. But I must admit that this part is not quite clear to me and these are my questions about it:

  • (1) Breeding … - first sentence is not clear. I think authors should rewrite complete paragraph to be clear and understandable. What does it mean “… at home and abroad…”? What does it mean “active validity period and stability”? Above all, what does it mean “disease-resistant AMF strains”? As much as I understand, the aim of this point could be to cultivate and construct the bank of AMF strains that cause beneficial processes in plants when plants face pathogenic attack in order to make plant resistance, not to collect AMF strains that are resistant to diseases. This sound strange, maybe even incorrect. Authors also mention possibility of extraction of the antibacterial substances from disease-resistant AMF strains – if such studies are published, this should be described somewhere in the text or authors should explain on what basis do they recommend this. From the whole this paragraph I would keep only the last two sentences in this form.

Also the subtitle “Breeding, domestication and construction of AMF strains with strong disease resistance” is problematic to me – I am not sure that breeding is suitable term (I would rather say cultivation). “Construction of AMF strains with strong disease resistance” sounds like genetic engineering with the goal of production of some super AMF strains. Maybe authors wanted to say “Construction of bank of AMF strains with strong potential to enhance plant hosts’ disease resistance”? – this would be clear to me.

  • (2) I think authors should write a subtitle or some paragraph in the maintext about combined action of AMF and other beneficial microorganisms. There are numerous studies on this topic published and it is not something that is only a future need.
  • (4) Throughout the manuscript text authors described numerous examples how AMF may activate some physiological processes that enhance plant immunity system (hormones, enzymes, defence gene expression etc.). And many of such studies refer to above ground plant. So, I don’t see why should “aerial parts of the plants” be mentioned as some future focus when this is already the case. I would omit this.
  • The entire “Conclusions and prospects” part is insufficiently well written according to me. I suggest writing only main beneficial influences of AMF on control of plant disease and some main future prospects, from which I definitely advise to omit majority of the point (1), moving point (2) to main text, omit point (4) and to shorten point (5) and rewrite it in a clearer way.

Other comments:

  • Line 68, 2. Biocontrol of AMF against phytopathogenic fungi – the writing style is mainly listing of what different researchers have found which is difficult to read. I think it should be written more fluid to be easier to follow.
  • Line 98, 3. Biocontrol of AMF against phytopathogenic bacteria and nematodes - The same thing about writing style as in subtitle 2. Moreover, too small part refers to AMF influence on pathogenic bacteria.
  • Line 126, 4. Plant nutrition – there are numerous papers published on the influence of AMF on plant nutrition in various plant species. This is why I think authors should put much more attention to this topic and write it more complete, not only generally. I suggest to move this subtitle into subsubtitle of the point 5. (The mechanism of AMF improving plant disease resistance). Also, lines 133-138 – do they better suit to the part about competition? Figure 1 – I think this figure is not suitable to the part of AMF influence on plant nutrition, it is not informative enough.
  • Line 259 – “changing the physical and chemical properties of the plant rhizosphere soil” is not described, even mentioned anywhere in the manuscript. So this can not be conclusion here.
  • Lines 275-277 – I don’t think this belongs to this subtitle.
  • Line 262 - relation between AMF and viruses is not mentioned anywhere in the manuscript (omit it from the Abstract).

Author Response

The manuscript is difficult to follow since lots of information is incorporated in the text and the style should be more fluent and less listing-like.

Specific comments:

Line61: More reference should be added to confirm this statement.

We have inserted more references to confirm this statement.

Line 175: Reference 56 is not about the resistance of host plants to Verticillium dahlia.

We have provided a new reference.

Line 178: Reference 57 is not about hindering pathogen penetration into root cell tissues and further infection upom mycorrhization.

We have provided a new reference.

Line 178: statement that changes in root anatomy alter the infection kinetics of pathogens. Not sure this is correct so please write kinetics of which pathogen and provide a reference.

We have rewritten this sentence. We have provided a reference.

Line 181: Please provide a reference.

We have provided a reference.

Line 238: “Studies have shown that mycorrhizae and their extra-root hyphae can penetrate through the tiny pores between soil particles, and their secretions, such as glomus associated protein (GRSP), organic acids, and polyamines,  can be used as adsorption agents for the adhesion of soil particles, promote the formation  of soil aggregate structures, improve soil pH, water stability, aeration, and water permeability, further increase the redox potential (Eh), and promote normal plant growth to resist disease invasion [72].”Please insert more references to confirm your statements.

We have inserted more references.

Line 267: “After AMF infestation of plant roots, the number of invasion sites of parasitic pathogens was significantly reduced [73].” Please rewrite and insert which plant and which pathogen.

We have rewritten this sentence.

Reviewer 3 Report

The manuscript details the importance of the effects of mycorrhizal fungi binding to plants. The manuscript needs intensive revision for grammatical and journal standard errors.

The bibliography does not follow publication standards.

Examples:

Line 92-95: Steinkellner et al. tested tomatoes with G. intraradices and the pathogen F. ox- 92 ysporum and showed that the severity of the disease could be reduced using a combination 93 of phosphorus application and AM fungal pre-treatment [25]. Few studies have reported on 94 the effects of AMF on airborne fungal diseases.

[25]  should go behind the reference:  Steinkellner et al. [25]  tested …..

LINE 100, 175  P. solanacearum  or Verticillium must be in cursive. review all scientific names

Line 213; systems of plants [64].Flavonoids can also increase the invasion sites of AMF and increase: Need space

Author Response

The manuscript details the importance of the effects of mycorrhizal fungi binding to plants. The manuscript needs intensive revision for grammatical and journal standard errors.

A lot of grammatical and journal standard errors of the manuscript had been revised.

The bibliography does not follow publication standards.

The bibliography has been revised to comply with the journal's publication standards.

Examples:

Line 92-95: Steinkellner et al. tested tomatoes with G. intraradices and the pathogen F. ox- 92 ysporum and showed that the severity of the disease could be reduced using a combination 93 of phosphorus application and AM fungal pre-treatment [25]. Few studies have reported on 94 the effects of AMF on airborne fungal diseases.

[25] should go behind the reference: Steinkellner et al. [25] tested …..

[25] has been placed after Steinkellner et al.

Line 100, 175 P. solanacearum or Verticillium must be in cursive. review all scientific names

Line 100, 175 P. solanacearum or Verticillium has been presented in cursive form. We also checked and revised all scientific names in the manuscript.

Line 213; systems of plants [64]. Flavonoids can also increase the invasion sites of AMF and increase: Need space

A space has been added between [64]. and Flavonoids.
